# Extracellular Vesicles and Carried miRNAs in the Progression of Renal Cell Carcinoma

**DOI:** 10.3390/ijms20081832

**Published:** 2019-04-13

**Authors:** Cristina Grange, Alessia Brossa, Benedetta Bussolati

**Affiliations:** 1Department of Medical Sciences, University of Turin, via Nizza 52, 10126 Turin, Italy; cristina.grange@unito.it; 2Molecular Biotechnology Centre, University of Turin, via Nizza 52, 10126 Turin, Italy; alessia.brossa@unito.it; 3Department of Molecular Biotechnology and Health Sciences, University of Turin, via Nizza 52, 10126 Turin, Italy

**Keywords:** exosomes, renal cancer stem cells, microRNAs, metastasis

## Abstract

The formation and maintenance of renal cell carcinomas (RCC) involve many cell types, such as cancer stem and differentiated cells, endothelial cells, fibroblasts and immune cells. These all contribute to the creation of a favorable tumor microenvironment to promote tumor growth and metastasis. Extracellular vesicles (EVs) are considered to be efficient messengers that facilitate the exchange of information within the different tumor cell types. Indeed, tumor EVs display features of their originating cells and force recipient cells towards a pro-tumorigenic phenotype. This review summarizes the recent knowledge related to the biological role of EVs, shed by renal tumor cells and renal cancer stem cells in different aspects of RCC progression, such as angiogenesis, immune escape and tumor growth. Moreover, a specific role for renal cancer stem cell derived EVs is described in the formation of the pre-metastatic niche. We also highlight the tumor EV cargo, especially the oncogenic miRNAs, which are involved in these processes. Finally, the circulating miRNAs appear to be a promising source of biomarkers in RCC.

## 1. Introduction

### 1.1. Renal Cell Carcinoma

Renal cell carcinoma (RCC) accounts for about 3% of all adult malignancies, being the twelfth most common cancer in the world [1] and the third most common urogenital malignancy [2,3]. RCC has the highest incidence in males and is one of the fastest increasing cancers, with this trend expected to continue over the next 20 years [4]. 

Although different histological subtypes of RCC are described, clear-cell RCC occurs most frequently and accounts for up to 80% of the RCC new cases. Clear-cell RCC is histologically characterized by the presence of cancer cells with a transparent cytoplasm, which is due to the accumulation of cholesterol esters, phospholipids and glycogen, and a well-defined cell membrane [5]. The other subtypes are papillary, chromophobe RCC and collecting-duct carcinoma. Papillary RCC (15% of RCC) is the principal cancer type in kidney transplant recipients while chromophobe RCC, which has the best prognosis, is quite rare [6]. 

Numerous genetic mutations are known to be involved in the pathogenesis and progression of RCC and their identification would contribute to better diagnoses and prognoses [7]. This is crucial in the development of new specific anti-cancer therapeutic strategies. The most common genetic abnormality and the first described is the inactivation of the tumor suppressor von Hippel-Lindau (VHL) by mutations, loss of heterozygosity or promoter hypermethylation [8]. The VHL protein is part of an E3 ubiquitin ligase multi-protein complex that regulates protein degradation through proteasomes [9]. A loss of function in VHL generates an upregulation of hypoxic inducible factors (HIF)-1α and 2α, which heterodimerize and stimulate the transcription of pro-angiogenic proteins, namely vascular endothelial growth factor (VEGF) and platelet-derived growth factor (PDGF) [10,11]. In particular, the activation of VEGF related pathways stimulates the proliferation, migration and survival of endothelial cells. This genetic mutation occurs mainly in the clear-cell RCC subtype. However, the inactivation of VHL per se is not sufficient to trigger RCC [1,10]. Other mutations have been described to contribute to RCC initiation and progression, such as SWI/SNF chromatin-remodeling complex gene PBRM1, BRCA1 associated protein-1, SET domain containing 2 and lysine K-specific demethylase 6A [12]. Moreover, it has been shown that the mammalian target of rapamycin (mTOR) pathway is significantly increased in RCC, which has a role in cell growth regulation in response to hypoxia [13]. 

Several studies have recently analyzed the microRNA (miRNA) expression profiles of RCC tissue specimens and have described an upregulation of miRNAs that target tumor-suppressors along with a downregulation of miRNAs that target oncogenes [14,15]. Deregulated miRNAs influence key molecules that are implicated in RCC progression, such as PTEN, VHL, HIF, VEGF and mTOR [16]. RCC is characterized by poor prognosis due to its high metastasis rate and difficulty in diagnosis. In fact, over 60% of RCC are detected incidentally. Despite the improvement of imaging techniques, about 20–30% of all patients at the time of diagnosis are already found to have metastatic cancer [1] and about 30% of patients treated for localized RCC have a relapse in distant sites [17,18]. The prognosis of patients with metastatic RCC is extremely poor with a 5-year survival rate of less than 10% [19,20]. The incomplete eradication of tumor cells is one of the factors of treatment failure and this may be due to cellular heterogeneity. In particular, the presence of a small population of cancer stem cell (CSCs) is attracting interest in the field as the major cause of tumor recurrence and resistance to therapy [21,22].

### 1.2. Renal Cancer Stem Cells

It is well established that the heterogeneity of solid tumors requires the presence of a subpopulation of cancer cells with a stem-like phenotype, which are the so-called cancer stem cells. Their main feature is their ability to differentiate into different cell types and sustain tumor growth [23]. This population resides in a well-defined tumor area that participates in the maintenance of stem phenotype, which is called niche, where CSCs and non-malignant cells may interact and influence each other [24]. Renal CSCs have been distinguished in RCC by stem cell marker isolation or by functional characteristics, such as sphere formation or dye exclusion [25]. In the first report in 2008, renal CSCs were identified as a population expressing Endoglin (CD105) and other markers of stemness, such as Nestin, Nanog and Oct-4 [22]. Renal CSCs have been shown to possess tumor initiating features and to differentiate towards epithelial or endothelial phenotypes both in vitro and in vivo. CD105, a TGF-β co-receptor, is expressed by many cell types, including mesenchymal and endothelial cells, and its expression is correlated with poor prognosis in RCC [26]. Recently, a high expression of CXCR4 was proposed to be a new CSC marker that correlates with more aggressive and metastatic RCC [27]. As an alternative, Addla et al. defined renal CSC as the side population (SP), which has the characteristic of Hoechst 33342 dye exclusion [28]. The renal SP consists of cells with a high proliferation rate and stemness properties [28]. Moreover, the sphere formation capacity was used to select renal CSCs in human SK-RC-42 RCC cell line [29].

The origin of renal CSCs is controversial as they may derive from the transdifferentiation of normal renal stem cells with mesenchymal phenotype or conversely from dedifferentiation of renal carcinoma cells. Moreover, renal CSCs may participate in different steps of metastatic tumor progression by sustaining tumor vascularization and the formation of a pre-metastatic niche, which possibly occurs through a long range signaling network [30]. Among the mechanisms of communication between primary tumor cells and other cellular components of the pre-metastatic lesion, many studies focused their attention on the emerging role of tumor-derived extracellular vesicles (EVs) and their miRNA cargo.

## 2. Effects of RCC-Derived EVs

### 2.1. Tumor EVs

The term “extracellular vesicles”, recommended by the International Society of Extracellular Vesicles (ISEV), defines a mixed population of vesicles with different dimensions that are released by most cell types. These vesicles are classically divided into exosomes and microvesicles depending on their biogenesis [31]. Based on recent studies, it is well recognized that EVs orchestrate both physiological and pathological cell-to-cell communication through the delivery of cellular components from the cell of origin to target cells. EVs have been isolated from the majority of biological fluids and due to their stability in circulation, they could be a suitable diagnostic tool [32]. 

Several non-tumor cells, such as fibroblasts, endothelial cells and immune cells, participate in the establishment of a favorable microenvironment to promote tumor progression and metastatization [33]. This phenomenon has been partially attributed to EV secretion [34]. In cancer patients, the number of circulating tumor-derived EVs has been correlated with poor prognosis [35]. Indeed, EVs released by tumor cells mediate different steps of tumor progression by the transfer of oncogenic molecules, such as lipids, proteins and nucleic acids [36]. Apart from the fact that all tumor EVs trigger tumor progression, it can be speculated that EVs released by CSCs may possess peculiar properties due to their specific cargo. Among all the factors that are carried by RCC-EVs, a major role was ascribed to miRNA content [32]. EV miRNA cargo has been shown to be involved in the regulation of multiple targets in recipient cells, triggering all the pivotal processes during tumor progression.

In this review, we will dissect the different aspects of RCC-derived EVs in tumor progression and metastatization. In particular, we will analyze the local effects of RCC-EVs within the tumor bulk, acting on tumor cells and other non-tumor cells, such as mesenchymal and endothelial cells. In addition, the systemic effects of RCC circulating EVs on immune cells and their action in the formation of premetastatic niche will be discussed. Finally, we will report the EV cargo that is responsible for the observed effects, including proteins, miRNAs and mRNAs (Figure 1).

### 2.2. Pro-Angiogenic Ability of RCC-EVs

Tumor EVs have been extensively described to promote angiogenesis by carrying pro-angiogenic mRNAs and miRNAs and modulating angiogenic pathways [37,38]. Several miRNAs, such as miR-23a, miR-210, miR-135b, miR-494, miR-1246 and miR-9, carried by tumor EVs have been demonstrated to be transferred to endothelial cells that display pro-angiogenic activity [39]. In particular, in lung carcinoma, Hsu et al. have described that hypoxic cancer cells release EVs that are enriched with miR-23a, which subsequently generate the accumulation of HIF-1α in endothelial cells and promotes angiogenesis [40]. miR-23a directly suppresses prolyl hydroxylase 1 and 2 (PHD1 and 2), which hydroxylates HIF-1α and facilitates its degradation through proteasomes [40]. Subsequently, the presence of a high level of miR-23a favors the accumulation of HIF-1α in endothelial cells, resulting in cell proliferation and migration [40]. Moreover, it has been described that under hypoxia, tumor cells increase the release of exosomal miR-210 [41,42]. miR-210 targets receptor tyrosine kinase ligand ephrin A3 and phospho-tyrosine phosphatase 1B, both negative regulators of VEGF, which causes angiogenesis stimulation [39,42]. 

In RCC, hypoxic conditions stimulate the release of tumor cells-derived EVs that are enriched with carbonic anhydrase 9 (CAIX), an HIF induced gene that regulates the proliferation and migration of endothelial cells [43]. CAIX is a transmembrane protein overexpressed in several solid tumors, which is involved in the regulation of intracellular pH and improves cancer cell survival [43,44]. In addition, RCC-EVs are enriched with the azurocidin protein (AZU1) that modulates vascular permeability and affects metastatization [45]. Specific pro-angiogenic and pro-tumorigenic roles have been attributed to EVs released by renal CSCs [30,32]. In fact, EVs released by CSCs and undifferentiated tumor cells carry several pro-angiogenic mRNAs, such as VEGF, FGF, angiopoietin 1, ephrin A3, MMP2 and MMP9. These are all involved in the stimulation of endothelial cell growth, invasion and resistance to apoptosis, which favors tumor progression and angiogenesis [30] (Figure 1). 

### 2.3. Effect of RCC-EVs on Tumor Stroma

Another important factor that is necessary to stimulate tumor progression is the interaction of tumor with stromal cells, especially mesenchymal stromal cells (MSCs) [46]. The effect of resident MSCs on tumor progression is not well established and both pro-tumorigenic and anti-tumorigenic activity have been reported [47]. This could be due to the effect of tumor secretoma that influences MSCs towards different and even opposite characteristics. Lindoso et al. described that MSCs treated with renal CSC-derived EVs were able to sustain tumor progression by enhancing MMP1, MMP2, MMP3, collagen 4A3, CXCR4 and CXCR7 expression levels, which results in increased tumor vascularization and proliferation [46]. MMPs are well known to regulate angiogenesis and tumor invasion and metastasis [48] while CXCR4 is involved in MSC migration [49] and CXCR7 in MSC survival [50].

In addition, mesenchymal cells derived from bone marrow (BMDCs) are also involved in the generation of an advantageous tumor microenvironment that leads to metastasis development. Recently, it has been demonstrated that the exosomes released by melanoma cells influence BMDCs toward a pro-vasculogenic and pro-metastatic phenotype. Tyrosinase-related protein-2, very late antigen 4, heat shock protein 70 and MET oncoprotein were all found within exosomes, with the MET oncogene being the principal molecule responsible for triggering the transformation of BMDCs [51].

### 2.4. Immunosuppressive Effects of RCC-EVs 

Recent studies have proposed that tumour cells influence the immune response and inhibit T cell proliferation, natural killer (NK) activation and dendritic cell (DC) differentiation through the delivery of EV cargo [52,53]. Focusing on RCC, Grange et al. demonstrated that CSC-derived EVs isolated from clear-cell RCC are the main mediators for the inhibition of DC differentiation and maturation from monocytes [54]. In particular, DCs that differentiated in the presence of renal CSC EVs expressed a low level of costimulatory molecules CD80 and CD86; other activation markers, such as CD83 and CD40; and the antigen-presenting molecule HLA-DR. Moreover, DCs that are differentiated in the presence of renal CSC EVs were unable to induce T lymphocyte proliferation and they released anti-inflammatory cytokines, such as IL-10 [54]. One of the main players of this inhibitory effect was the non-classical HLA class I molecule, HLA-G. HLA-G was previously described to abrogate NK, T cell and DC functions [55]. In addition, HLA-G has been found to be overexpressed in 50% of clear-cell RCCs [56]. Moreover, EVs isolated from human kidney adenocarcinoma were able to induce T lymphocyte apoptosis through the activity of FAS ligand expressed on their surface [57] (Figure 1). 

### 2.5. Effect of RCC-EVs on the Premetastatic Niche

The metastatic niche is defined as a number of molecular and cellular changes in the metastatic designated organs that favor tumor cell colonization and sustain tumor establishment in a distant site [58]. All the components of secretomes, including soluble factors, EVs, exosomes and microvesicles, support this process, creating a complex network of information exchanges [59,60]. 

A crucial step to initiate the process for tumor cell dissemination from a primary to foreign site is the stimulation of vascular permeability [39]. Zeng et al. demonstrated that miR-25-3p, released by colorectal cancer exosomes, is transferred to tumor endothelial cells, which results in increased vascular permeability and consequently increased metastasis formation. miR-25-3p targets Krüppel-like factor 2 (KLF2) and Krüppel-like factor 4 (KLF4), a family of zinc finger-containing transcription factors [39]. KLF2 inhibits angiogenesis by reducing the activity of VEGFR2 while KLF4 is involved in the maintenance of endothelial barrier integrity and acts on tight junction proteins, such as ZO-1, occludin and claudin 5 [61]. 

In RCC and prostate cancer, the ability to promote the premetastatic niche formation was mainly retained by CSCs EVs rather than by those derived from the tumor bulk [30,62]. In fact, EVs released by renal CSCs and not by differentiated tumor cells enhance the formation of lung metastases by the upregulation of MMP2, MMP9 and VEGF receptors by lung endothelial cells [30]. Comparing the miRNA profile of the EV populations released by CSCs and differentiated tumor cells, it has been shown that renal CSC EVs were enriched with miRNAs that may influence biological pathways related to cell growth, cell matrix adhesion and survival [30]. In particular, these EVs shuttled several miRNAs, such as miR-19a, miR-19b, miR-650 and miR-151, which were previously associated with tumor invasion and metastases (Table 1) [63,64,65]. Recently, a similar approach was applied by Sanchez et al. for the comparative analysis of miRNA content in prostate cancer and prostate CSC EVs using next generation sequencing [62]. miR-100-5p and miR-21-5p have been described as the most abundant miRNAs while the differentially expressed miRNAs were predicted to be involved in prostate carcinogenesis, fibroblast proliferation, differentiation and angiogenesis [62]. The overexpression of miRNA-183 was found in EVs that were shed by renal and prostate CSCs [30,62] (Figure 1).

### 2.6. RCC-EV oncomiRNAs

miRNAs are 19–22 bp non-coding RNAs involved in the regulation of biological processes by the post-transcriptional downregulation of one or more target genes. MiRNAs bind to a complementary sequence that is present in target mRNA, which leads to its degradation [66]. To date, the number of miRNAs annotated in human genome is continuously increasing and it can be speculated that the majority of protein-coding genes are targeted by at least one miRNA [67]. In the last few years, numerous reports have shown the importance of miRNAs in the regulation of cancer related biological processes [68]. During tumor progression, miRNAs can act as oncogene or tumor suppressors depending on their mRNA targets and on the cellular context. All the miRNAs that inhibit tumor suppressor proteins involved in relevant processes, such as cell cycle, apoptosis, DNA damage and chemoresistance, are generally defined as “oncomiRNAs” [69,70]. Several oncomiRNAs have been identified in renal CSC EVs and are discussed above for their pro-tumorigenic role (Table 1) [30].

#### 2.6.1. miR-183

miR-183 belongs to the miR-183-96-182 cluster, which has an upregulated expression in many solid tumors, including RCC [72]. The expression of miR-183 was increased in tumor specimens compared to their healthy counterparts and its role in promoting invasion and metastasis was demonstrated in RCC lines [80]. Indeed, high levels of miR-183 were also detected in the sera from RCC patients and was inversely correlated with the cytotoxicity of NKs compared to tumor cells in vitro, thus suggesting a possible application of miR-183 detection in patients’ sera as a prognostic marker for immunotherapy [80]. Finally, Li et al. analysed the sera from 284 RCC patients that underwent nephrectomy and demonstrated a positive correlation between miR-183 level and poor prognosis [81]. miR-183 promotes tumor cell proliferation and invasion by targeting Dickkopf-related protein-3 (DKK-3), a negative regulator of the Wnt signaling pathway [80], and the tumor suppressor protein phosphatase 2A, one of the main regulators of AKT signaling [71]. 

#### 2.6.2. miR-486

Goto et al. analysed formalin-fixed paraffin-embedded specimens from RCC patients and identified miR-486 as a possible biomarker for poor prognosis [74]. Indeed, He et al. recently demonstrated the role of miR-486 in promoting proliferation and inhibiting apoptosis of RCC cell lines in vitro [73]. miR-486 has been proposed to be a negative regulator of multiple targets involved in PTEN/PI3K/Akt, FOXO and TGF-b/Smad2 signaling [82].

#### 2.6.3. miR-92, miR-19a and miR-19b

miR-92, miR-19a and miR-19b are components of the miR-17-92 cluster, which is also called Oncomir-1. The cluster consists of six miRNAs, namely miR-17, miR-18a, miR-19a, miR-20a, miR-19b and miR-92a, and was found to be upregulated in many solid tumors, including RCC [79,83]. The components of the miR-17-92 cluster target the genes involved in proliferation and survival, thus inhibiting apoptosis and increasing cell proliferation, invasion and escape from immune system [84]. In RCC, miR-92 targets the tumor suppressor gene VHL, which allows the stabilization of HIFs, a crucial point during RCC progression as discussed above [78]. miR-19a, targeting PTEN [75] and RhoB [77], influences both proliferation and invasiveness of RCC cells. Finally, miR-19a has been described to be overexpressed in RCC tissues and especially has been shown to be upregulated in metastatic RCC compared to primary tumor tissues [75]. For these reasons, miR-19a has been proposed as a possible prognostic biomarker for RCC [76].

#### 2.6.4. miR-650

miR-650 has been described as an oncomiRNA involved in the metastatic progression of different solid tumors, such as gastric cancer [64], prostate cancer [85], colorectal cancer [86], hepatocellular carcinoma [87] and oral cancer [88]. Although the presence of high levels of miR-650 has been reported in renal CSC EVs, the role of this oncomiRNA in RCC progression is not yet fully understood.

#### 2.6.5. miR-301

An in vitro study on breast cancer cells described miR-301 as one of the central oncomiRNAs in breast cancer [89], with miR-301 being involved in many steps of tumorigenesis through multi-target activity and its expression was associated with the presence of distant metastasis [90]. miR-301 was also detected in both small cell carcinoma specimens [91] and in EVs isolated from non small cell lung cancer patients [92]. Although both authors proposed miR-301 as a possible biomarker for lung cancer, the study of Silva et al. failed to demonstrate that this miRNA could discriminate between healthy and cancer patients [92].

### 2.7. Long Non-Coding RNAs in RCC-EVs

It is increasingly evident that in many types of EVs, other forms of non-coding RNAs, such as long non-coding RNAs (lncRNA), are more represented than miRNAs [93]. LncRNAs are a class of transcripts with a length of 200 bp or longer than regulate gene expression at various levels. Indeed, lncRNAs can control cell transcription and protein translation by interacting with proteins, mRNAs or miRNAs [94]. LncRNAs have been reported to regulate different steps of tumorigenesis, such as cell proliferation, apoptosis and invasion, and may also have diagnostic value [95]. Interestingly, a recent study identified an lncRNA, namely lncARSR (for lncRNA Activated in RCC with Sunitinib Resistance), in RCC tissue and the tumor released EVs [96]. LncARSR promoted Sunitinib resistance through the regulation of miR-34/miR-449, thus facilitating AXL and c-MET expression in RCC cells. Moreover, the intracellular transfer of lncARSR by RCC-EVs disseminated resistance to Sunitinib and targeting this could restore the drug response. The role of the lnc-RNAs is of great interest in tumor EV biology although further studies are needed to identify EV- and circulating-lncRNAs and to better dissect their role in RCC and RCC-derived EVs.

## 3. Circulating miRNAs in RCC

In the last decade, it has been discovered that miRNAs are present at high levels in the bloodstream of cancer patients [35]. Circulating miRNAs are ideal candidates for diagnostic purposes since they can be isolated in a non-invasive manner from biologic fluids. miRNA stability depends on the presence of circulating RNAse and it is increased when miRNAs are bound to lipoproteins or encapsulated in EVs. Many studies were recently focused on the identification of miRNAs as possible diagnostic and prognostic biomarkers for RCC [97] (Table 2).

In 2011, Wulfken et al. published the first study that aimed to identify the miRNA signature in the sera of RCC patients [97]. The analysis of blood samples from 190 patients (93 healthy and 97 cancer patients) identified 36 miRNAs that were differentially expressed between healthy people’s and patients’ sera. Among them, only miR-1233 was validated as a possible biomarker for diagnostic purposes [97].

Other miRNAs were subsequently identified in patients’ sera and were proposed as candidates for RCC diagnosis as potential biomarkers in RCC [98,99,100,101,102,103,104]. In particular, a large-scale analysis on both healthy subjects and cancer patients before and after nephrectomy highlighted the combination of miR-210 and miR-378 as potential biomarkers for early tumor detection [105].

As shown in Table 2, each study identified a different subset of circulating miRNAs, highlighting the absence of correspondence among the studies. Indeed, a genome-wide screening on circulating miRNAs in both early and late stage renal carcinoma patients detected a subset of 27 miRNAs that were differentially expressed between early and late stage patients [106]. However, authors do not support the use of circulating miRNA as diagnostic or prognostic biomarkers since the expression of the identified miRNAs were not significantly correlated with tumor progression. 

## 4. Conclusions

The crosstalk between all cell types within the tumor, including non-malignant cells, is pivotal in tumor maintenance. RCC-EVs and even more renal CSC-EVs sustain tumor growth by acting on several aspects of renal tumor progression, such as tumor proliferation, angiogenesis, immunoescape and metastasis formation. The complexity of the EV cargo (mRNAs, proteins and especially miRNAs) is the basis for the multi-target activity described above, involved in the reprogramming of target cells. Therefore, the development of strategies to interfere with EV mediated intercellular communication is now considered to be a challenging therapeutic option. In addition, tumor EVs have been proposed as new biomarkers for tumor diagnosis due to their high amount in biological fluids and increased stability compared to non-EV encapsulated molecules. However, in RCC, the identification of a prognostic miRNA still requires further standardization and validation.

## Figures and Tables

**Figure 1 ijms-20-01832-f001:**
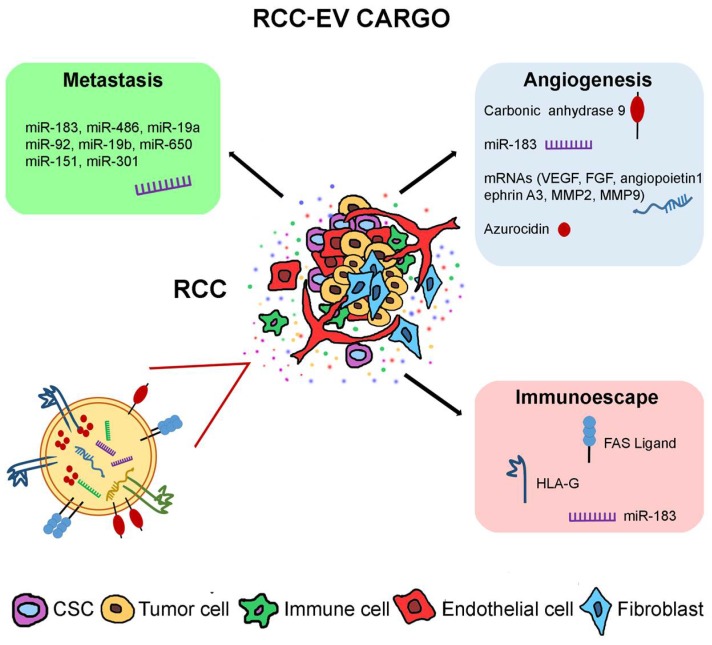
RCC-EV cargo. Molecules carried by RCC-EVs are described to play a role in tumor progression through the promotion of angiogenesis, metastasis formation and immune escape.

**Table 1 ijms-20-01832-t001:** OncomiRNAs carried by renal CSC EVs. The table lists some of the oncomiRNAs carried by renal CSC EVs and highlights their targets in renal cell carcinoma and the renal samples used to identify specific miRNAs.

miRNAs	Targets in RCC	Renal Samples	Reference
**miR-183**	Dickkopf-related protein-3 (DKK-3) Protein phosphatase 2A	Renal CSC EVs Renal tumor cells Renal cancer tissues Serum of RCC patients	Grange et al. [30] Qiu et al. [71] Zhang et al. [72]
**miR-486**	TGF-β-activated kinase 1	Renal CSC EVs Renal tumor cells Renal cancer tissues	Grange et al. [30] He et al. [73] Goto et al. [74]
**miR-19a**	PTEN SMAD4 RhoB PIK3CA	Renal CSC EVs Renal cancer tissues	Grange et al. [30] Ma et al. [75] Xiao et al. [76] Niu et al. [77]
**miR-92**	VHL	Renal CSC EVs Renal cancer tissues	Valera et al. [78] Grange et al. [30]
**miR-19b**	RhoB PTEN/PI3K/AKT Signalling Pathway	Renal CSC EVs	Grange et al. [30] Niu et al. [77] Liu et al. [79]
**miR-650**	Not determined	Renal CSC EVs	Grange et al. [30]
**miR-151**	Not determined	Renal CSC EVs	Grange et al. [30]
**miR-301**	Not determined	Renal CSC EVs	Grange et al. [30]

**Table 2 ijms-20-01832-t002:** Circulating miRNAs in RCC patients. The table lists some of the circulating miRNAs isolated in serum of RCC patients and specifies the screening technique, screening groups and results.

miRNA	Screening Technique	miRNA Source	Screening Groups	Validation Groups	Study Results	References
miR-378	Real time PCR	Serum	25 healthy controls 25 RCC	123 healthy controls 117 RCC	miR-378 levels were higher in RCC screening group versus healthy controls, but results were not validated.	Hauser et al. [104]
	Real time PCR	Serum	15 healthy controls 12 RCC	35 healthy controls 90 RCC	Validation of the increase of miR-378 in patients’ sera, in combination with the downregulation of the antitumor miR-451.	Redova et al. [98]
	Real time PCR	Serum	100 healthy controls 195 RCC	Not performed	Diagnostic and prognostic value of miR-378 increase, in combination with miR-210 increase.	Fedorko et al. [105]
miR-221	Real time PCR	Plasma	34 healthy controls 44 RCC	Not performed	miR-221 levels positively correlate with progression towards a metastatic disease and inversely correlate with survival rate.	Teixeira et al. [101]
miR-210	Real time PCR	Tissue samples and serum	32 RCC tissues samples (tumor and adjacent healthy tissue); sera from 42 healthy controls, 68 RCC before and 10 RCC patients after nephrectomy	Not performed	Increased levels of miR-210 both in tissue samples (compared to adjacent healthy tissue) and in RCC sera.	Zhao et al. [99]
	Real time PCR	Tissue samples and serum	23 healthy controls 34 RCC	Not performed	Increased levels of miR-210 in RCC sera compared to controls. No correlation with tumor stage.	Iwamoto et al. [100]
	Real time PCR	Serum	100 healthy controls 195 RCC	Not performed	Increased levels of miR-210 in RCC sera compared to controls. Prognostic value in combination with miR-378.	Fedorko et al. [105]
	Real time PCR	Serum EV	80 healthy controls, 82 RCC before nephrectomy 10 RCC after nephrectomy.	Not performed	Higher content in miR-210 and miR-1233 in EV isolated from RCC patients’ sera.	Zhang et al. [107]
	Meta analysis	Serum	Not performed	7 studies: 415 healthy controls 570 RCC	Validation of the potential diagnostic value of miR-210.	Chen et al. [108]
miR-193a	TaqMan Low Density Array	Serum	25 healthy controls 25 RCC	107 healthy controls 107 RCC	Increased in RCC sera.	Wang et al. [102]
miR-362	TaqMan Low Density Array	Serum	25 healthy controls 25 RCC	107 healthy controls 107 RCC	Increased in RCC sera.	Wang et al. [102]
miR-572	TaqMan Low Density Array	Serum	25 healthy controls 25 RCC	107 healthy controls 107 RCC	Increased in RCC sera.	Wang et al. [102]
miR-1233	TaqMan Low Density Array	Tissue samples and serum	6 healthy controls and 6 RCC	93 healthy controls and 84 RCC	miR-1233 was identified and validated as increased in RCC.	Wulfken et al. [97]
	Real Time PCR	Serum EPCAM+ EV	80 healthy controls, 82 RCC before nephrectomy 10 RCC after nephrectomy.	Not performed	Higher content in miR-210 and miR-1233 in EV isolated from RCC patients’ sera.	Zhang et al. [107]
miR-99b	Sequencing	Tissue samples	40 RCC (tumor and adjacent healthy tissue)	65 RCC	miR-99b levels positively correlated with tumor progression after TKI therapy	Lukamowicz et al. [109]
miR-21	Real Time PCR	Serum	30 healthy controls 30 RCC before and after nephrectomy	Not performed	Increased levels in RCC sera.	Tusong et al. [110]
miR-106a	Real Time PCR	Serum	30 healthy controls 30 RCC before and after nephrectomy	Not performed	Increased levels in RCC sera.	Tusong et al. [110]
miR-122-5p	small RNA sequencing	Serum	8 benign renal tumors 18 RCC	28 healthy controls 47 benign renal tumors 68 RCC	Correlation of miR-122 levels with metastatic progression and survival	Heinemann et al. [103]
miR-206	small RNA sequencing	Serum	8 benign renal tumors 18 RCC	28 healthy controls 47 benign renal tumors 68 RCC	Correlation of miR-122 levels with metastatic progression and survival	Heinemann et al. [103]

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
