# Peer review of "Extracellular Vesicles and Carried miRNAs in the Progression of Renal Cell Carcinoma"

_ijms, 2019, doi:10.3390/ijms20081832_

Reviewer 1 Report

The manuscript is well written and provides a comprehensive review on the involvement of EV encapsulated miRNA in RCC. I have no specific suggestion on the current version of the manuscript.

Author Response

Thank you for your comments.

Reviewer 2 Report

The review manuscript by Grange et al. is clearly written and informative and deserves publication.

I have only two points that I would like to see addressed by the authors:

A. Other forms of non-coding RNA beside miRNA in EVs have been generally neglected in the general literature, in part because of unclear function(s). It is clear now that in many types of EVs the other forms of non-coding RNA are more represented than miRNAs. Although the Review focuses on miRNAs, it is this Reviewer's opinion that somewhere in the Review the authors should state the potential importance of these other forms.

B. To improve the flow, several typos and grammatical errors should be corrected. The following are just some examples:

line 21 these instead of theses

line 28 males instead of male

line 66 It is instead of it's

line 79 alternative instead of alternatively

line 106 EVs instead of EV

line 124 have described instead of have been described

line 125 generate instead of generates

Author Response

We thank to the reviewer for the comments that help to improve our manuscript.

A.    Other forms of non-coding RNA beside miRNA in EVs have been generally neglected in the general literature, in part because of unclear function(s). It is clear now that in many types of EVs the other forms of non-coding RNA are more represented than miRNAs. Although the Review focuses on miRNAs, it is this Reviewer's opinion that somewhere in the Review the authors should state the potential importance of these other forms.

Answer. We thank to the reviewer to point out this concept and we added a paragraph on long non-coding RNA. See page 8 lines 255-268 and references 93-96.

B. To improve the flow, several typos and grammatical errors should be corrected. The following are just some examples:

line 21 these instead of theses

line 28 males instead of male

line 66 It is instead of it's

line 79 alternative instead of alternatively

line 106 EVs instead of EV

line 124 have described instead of have been described

line 125 generate instead of generate

Answer. We thank to the reviewer for this comment and we correct several typos and grammatical errors.

Reviewer 3 Report

This is a nice review.

Author Response

Thank you for your comment.